# Prediction of Preterm Labor from the Electrohysterogram Signals Based on Different Gestational Weeks

**DOI:** 10.3390/s23135965

**Published:** 2023-06-27

**Authors:** Somayeh Mohammadi Far, Matin Beiramvand, Mohammad Shahbakhti, Piotr Augustyniak

**Affiliations:** 1AGH University of Science and Technology, 30059 Krakow, Poland; august@agh.edu.pl; 2Faculty of Information Technology and Communication, Tampere University, 33100 Tampere, Finland; 3Biomedical Engineering Institute, Kaunas University of Technology, 51423 Kaunas, Lithuania

**Keywords:** preterm labor, EHG, pregnancy week, EMD, AdaBoost

## Abstract

Timely preterm labor prediction plays an important role for increasing the chance of neonate survival, the mother’s mental health, and reducing financial burdens imposed on the family. The objective of this study is to propose a method for the reliable prediction of preterm labor from the electrohysterogram (EHG) signals based on different pregnancy weeks. In this paper, EHG signals recorded from 300 subjects were split into 2 groups: (I) those with preterm and term labor EHG data that were recorded prior to the 26th week of pregnancy (referred to as the PE-TE group), and (II) those with preterm and term labor EHG data that were recorded after the 26th week of pregnancy (referred to as the PL-TL group). After decomposing each EHG signal into four intrinsic mode functions (IMFs) by empirical mode decomposition (EMD), several linear and nonlinear features were extracted. Then, a self-adaptive synthetic over-sampling method was used to balance the feature vector for each group. Finally, a feature selection method was performed and the prominent ones were fed to different classifiers for discriminating between term and preterm labor. For both groups, the AdaBoost classifier achieved the best results with a mean accuracy, sensitivity, specificity, and area under the curve (AUC) of 95%, 92%, 97%, and 0.99 for the PE-TE group and a mean accuracy, sensitivity, specificity, and AUC of 93%, 90%, 94%, and 0.98 for the PL-TL group. The similarity between the obtained results indicates the feasibility of the proposed method for the prediction of preterm labor based on different pregnancy weeks.

## 1. Introduction

Preterm labor is defined as delivering a baby prior to the end of the 37th week of pregnancy, which is the primary cause of newborn mortality [1]. Based on the World Health Organization’s report, more than 15 million neonates are delivered prematurely each year, and of that, almost 1 million die because of ensuing complications [2]. In the European Union, for instance, the preterm delivery rate is between 5 and 10% [3]. Although there are several factors that have been considered to be related to preterm labor such as anxiety, multiple pregnancies, abortion, and short cervical length [4,5], it is still not clear to what extent these factors are related to preterm delivery, because almost 50% of preterm births occurred without any of the aforementioned factors [6].

Even in the case of survival, a premature neonate may face several serious complications such as difficulties in breathing and vision problems due to underdeveloped organs [7]. Furthermore, preterm labor may also have adverse influences on maternal well-being due to the mother’s perception of her baby [8]. Moreover, the expenses related to preterm labor healthcare inflict financial hardship on both society and family as the price of such healthcare is five to ten times more than a term delivery [9]. Therefore, early detection of preterm labor, in conjunction with proper medical care to avoid this phenomenon [10], is of great importance for increasing the chance of neonate survival, the mother’s mental health, and reducing financial burdens.

To this end, several techniques have been presented for the prediction of preterm labor, such as a tocodynamometer [11], ultrasound [12], fetal fibronectin [13], and internal uterine pressure [14]. The tocodynamometer monitors the uterine contractions by a pressure transducer placed on the fundus area. Although it is a noninvasive method, its performance highly relies on the accurate positioning of the sensor [15]. The ultrasound is used to measure the cervical length; however, as already mentioned, preterm labor may happen without any changes in cervical length. Fetal fibronectin and internal uterine pressure have also shown to be promising for the prediction of preterm labor; however, they are both invasive. As an alternative, the electrohysterography (EHG) modality can be used which represents the electrical activity of the uterus collected from the abdominal surface of a pregnant woman [16]. Despite the aforementioned techniques, it is a noninvasive method that can be automated to reduce the requirement of human expertise and is thought of as a future long-term ambulatory telecare tool.

The Term–Preterm ElectroHysteroGram DataBase (TPEHG DB) is the most studied dataset for the prediction of preterm labor using EHG signals. It consists of EHG signals recorded from 300 subjects, of which 262 delivered to term [17]. The efficiency of an EHG signal analysis for the prediction of preterm labor has been widely evidenced in the literature using this database [18,19,20,21,22,23,24,25,26,27,28,29,30,31]. In general, the feature extraction strategies for the EHG signal analysis can be classified into three groups: linear, nonlinear, and propagated EHG signal-related features [32]. Linear features such as the root mean square (RMS) [33,34] and median frequency [35] have shown to be promising to characterize EHG signals for preterm labor prediction. In [9], the effectiveness of several nonlinear features such as different entropy and fractal dimensions is widely investigated. Regarding the propagated EHG signal-related features, several studies show the importance of propagation velocity to discriminate between the term- and preterm-related EHG signals, e.g., [27].

Furthermore, due to the nonstationary and nonlinear characteristics of EHG signals, several studies investigate the potential of nonstationary algorithms for the feature extraction. One of the widely employed methods is empirical mode decomposition (EMD) [36], which decomposes a signal into several intrinsic mode functions (IMFs) ordered from high- to low-frequency components. In [37], the Shannon entropy of the first ten decomposed IMFs is used to discriminate between term and preterm EHG signals and an area under the curve (AUC) of 0.98 is reported. In [38], several linear and nonlinear features from $IMF_{3}$ and $IMF_{6}$ of the decomposed EHG signals are analyzed to classify between the term and preterm EHG signals. By employing a balanced subset of EHG data (26 term and 26 preterm), an accuracy of 95.70% is obtained. In [39], the EHG signals are firstly decomposed into 11 IMFs and then each IMF is decomposed by wavelet packet decomposition to another 6 levels. After ranking the features, an accuracy of 96.25% is achieved. In another study [40], the feature extraction is performed on the second to ninth IMFs of the decomposed EHG signals and an accuracy of 98% is reported.

Although the time when the diagnosis is made is proven to play an important role in subsequent pregnancy care, a majority of the mentioned studies have not investigated the efficiency of the proposed methods in different pregnancy weeks. Indeed, the mentioned studies only consider the term and preterm labor prediction regardless of actual fetus maturity. However, an earlier prediction of preterm labor can provide more time for the physician to analyze the situation, i.e., prescribe proper medications and monitor the pregnant woman more frequently [41]. To the best of our knowledge, only few studies address the prediction of preterm labor based on different gestational weeks. Peng et al. [42] classify the EHG data into two groups: those which are recorded before the 26th week of gestation (PE-TE) and after the 26th week of gestation (PL-TL). By extracting 31 linear and nonlinear features, an accuracy of 92% and 93% for the PE-TE and PL-TL groups is achieved, respectively. Smrdel and Jager [43] extract the median frequency and sample entropy for both groups. After employing the data balancing method, 93 term and 93 preterm data for the PE-TE group and 57 term and 57 preterm data for the PL-TL group are used for the classification by quadratic discriminant analysis (QDA). The authors report an accuracy of 97% and 100% for the PE-TE and PL-TL groups, respectively. Ahmed et al. [44] extract multivariate multiscale fuzzy entropy from both groups and report an accuracy of 95% and 94% for the PE-TE and PL-TL groups, respectively. Jager et al. [35] use simultaneously recorded EHG and tocogram data for the prediction of preterm labor for the PE-TE group and report an accuracy of 100%.

Despite the promising results for the prediction of preterm labor using PE-TE and PL-TL groups, there are two issues that have not been addressed in the aforementioned studies. Firstly, the employment of nonlinear features such as sample entropy and fuzzy entropy, which require parameter setting before computation, may threaten the generality of the method as such tuning is performed experimentally and it is not clear how well it works for unseen data. Secondly, with an artificially balanced dataset, the reported sensitivity of the methods to the real preterm EHG data was not accurately investigated, i.e., in the case of a sensitivity of 90%, it is not clear how much of that missed 10% is related to the real preterm data.

The objective of this paper is to present a reliable method for predicting the preterm labor based on different pregnancy weeks. For this aim, we extract several parameter-free linear and nonlinear features from the EHG signals decomposed by EMD and then feed them to several classifiers for the final prediction. As the uterus contractions become more intense and frequent near the delivery, it can be expected that the EHG signal indicating preterm labor will exhibit stronger and more frequent contractions compared to term labor. In other words, the EHG signal associated with the preterm class will have a higher presence of high-frequency components. On the other hand, the EMD technique breaks down a signal into its frequency components from high to low, so we utilize only the initial four IMFs for the feature extraction. The motivation for using EMD over the other decomposition methods such as wavelet is that it does not require a predefined basis function. Furthermore, EMD does not assume any prior knowledge about the signal, such as stationarity or linearity. This characteristic makes EMD particularly useful when dealing with nonstationary signals such as EHG that often exhibit complex and unpredictable behavior.

## 2. Dataset

In this paper, we have employed a publicly available dataset called Term–Preterm ElectroHysteroGram DataBase (TPEHG DB) which consists of EHG signals recorded from 300 pregnant women. These data were recorded at the Department of Obstetrics and Gynecology of the University Medical Center Ljubljana from 1997 to 2005 [17]. As displayed in Figure 1, 4 electrodes were placed on the abdominal surface of the pregnant woman. Using these electrodes, three bi-polar EHG channels, i.e., $CH_{1}=E2-E1$, $CH_{2}=E2-E3$, and $CH_{3}=E4-E3$, were formed. The signals were sampled at 20 Hz and the duration of each measurement was almost 30 min.

Of these 300 EHG data, 262 were considered term and 38 were considered preterm labor. More specifically, these 300 measurements could be divided into 2 groups based on the gestational weeks, i.e., before and after the 26th week of pregnancy. As shown in Table 1, 162 measurements were recorded prior to the 26th week of pregnancy, of which 143 were considered to be term labor, and 138 measurements were recorded after the 26th week of pregnancy, of which 119 were considered to be term labor. The former group is referred to as preterm early and term early (PE-TE) and the latter group is referred to as preterm late and term late (PL-TL). The main challenge of using the TPEHG DB is the lack of balance between term and preterm cases. In fact, the recordings were taken prior to the delivery without knowing the labor would be term or preterm.

## 3. The Proposed Method

Figure 2 displays the block diagram of the proposed method for disctinction of term and preterm labor. The details of proposed method are explained in the subsections below.

### 3.1. Preprocessing

Before starting the analysis, preprocessing the EHG signals is a necessary step to reduce interference originating from power line noise, subject’s respiration, and fetal and maternal electrocardiogram [45]. As the most meaningful frequency components of EHG signals vary between 0 and 5 Hz, the signals are band-pass filtered in a range from 0.08 to 4 Hz using a fourth-order Butterworth filter. In order to avoid the transient effect of filtering, the first and last 5 min of each measurement are discarded. Thus, the analysis is performed on the remaining 20 min of the measurements. Figure 3 shows an example of the 60 s filtered EHG signals.

### 3.2. EHG Signal Decomposition

Through the sifting process, EMD decomposes the EHG signal x[n] into *L* number of IMFs and one residue component where the original EHG signal can be reconstructed as follows:(1)$$x\left[n\right]=\sum_{i=1}^{L}IMF_{i}\left[n\right]+r\left[n\right],$$

The first IMF is computed using the following steps:1.Detecting the local extrema of x[n];2.Synthesizing the upper and lower signal’s envelopes from the detected extrema using cubic spline;3.Forming the local mean signal, $m_{1}\left[n\right]$, by averaging the upper and lower signal’s envelopes;4.Subtracting the local mean signal $m_{1}\left[n\right]$ from the original signal x[n] to acquire the first possible IMF candidate $y_{1}\left[n\right]=x\left[n\right]-m_{1}\left[n\right]$.

Now, the $y_{1}\left[n\right]$ must fulfill two conditions:1.The number of zero-crossings and local extrema must either be equal or differ at most by one.2.The average value of the envelopes defined by the local maxima and minima is zero.

If the $y_{1}\left[n\right]$ does not satisfy the conditions above, it is treated as a new signal and the whole sifting process is performed on it once again. This procedure iterates *k* times until the IMF conditions are satisfied. After finding the first IMF $y_{k}\left[n\right]$, the residue signal $r_{1}\left[n\right]$ is calculated by subtracting it from the original signal:(2)$$r_{1}\left[n\right]=x\left[n\right]-y_{k}\left[n\right],$$

The second IMF is then extracted from the first residue signal using the previous steps. In this paper, we use the first four decomposed IMFs for the feature extraction.

### 3.3. Feature Extraction

As the intensity and frequency of uterus contractions increase near the delivery, it can be expected that EHG signal representing the preterm labor contains stronger and more frequent contractions than term ones. Consequently, features that characterize strength, energy, and complexity are well justified on physiological background. Therefore, we extract root mean square (RMS), mean Teagar–Kaiser energy (MTKE), wavelet log entropy (WLE), Shannon’s entropy (SE), Katz fractal dimension (KFD), and Hurst exponent (HE) from each IMF of the EHG signal. The RMS, which can represent the strength of a signal, is expressed as
(3)$$\mathrm{RMS}=\sqrt{\frac{1}{N}\sum_{n=1}^{N}x^{2}\left[n\right]},$$

The Teagar–Kaiser energy is a widely recognized tool for detecting the onsets of muscle contraction. It computes the energy of a signal as follows:(4)$$\mathrm{TKE}=x^{2}\left[n\right]-x\left[n-1\right]x\left[n+1\right],$$

As it is expected that EHG signal related to the preterm labor contains more contractions, we can assume that the average of TKE will be of discriminative power between term and preterm labor.

Entropy is used to quantify the uncertainty of a signal. Because EHG signal related to the preterm labor may have stronger and more frequent contractions, entropy measures can be proper index for the preterm–term EHG signal discrimination. Here, we use WLE and SE defined as follows [46]:(5)$$\mathrm{WLE}=\sum_{n}\log\left(x{\left[n\right]}^{2}\right),$$
(6)$$\mathrm{SE}=-\sum_{i=1}^{N}x\left[p_{i}\right]\times log_{2}\left(x\left[p_{i}\right]\right),$$
where x[n] and $p_{i}$ are the signal and the probability for obtaining the value $x_{i}$.

The KFD is used to quantify the signal’s crudity. It indeed represents the signal self-similarity. Let x[n] be a signal with n=1,2,⋯N, and the KFD is computed as
(7)$$\mathrm{KFD}=\frac{log(N-1)}{log\left(N-1\right)+log\left(\frac{D}{L}\right)},$$
where $L=\sum_{i=2}^{N}x_{i}-x_{i-1}$ is the total length of the curve and $D=Max(|x_{1}-x_{j}|)$ for j=2,3⋯N is the diameter of the waveform.

The HE measures the long-term memory of a signal as follows:(8)$$\mathrm{HE}=\frac{log(\frac{R}{S})}{log(N)},$$
where *N*, *R*, and *S* stand for the signal’s length, the difference between the maximum and minimum deviations from the mean, and the standard deviation, respectively.

### 3.4. Data Balancing

The occurrence of class imbalance is a common problem for medical diagnosis application, in particular for the minor classes which are of greater interest. For the dataset employed in this paper, almost 87% of data is related to the term labor. Several studies show that classifiers are biased toward the term labor class when using the original database, e.g., [47]. Thus, the employment of the data balancing algorithm is necessary for proper training of the classifiers. In this paper, we use the self-adaptive synthetic over-sampling (SASYNO) method [48], which is shown to be superior over the state-of-the-art data balancing methods. After extracting the features from each group, the SASYNO is applied on them which results in a equally distributed feature set for both term and preterm cases.

### 3.5. Feature Selection

In total, 72 features are extracted from the EHG data. To reduce the feature vector dimension, Mann–Whitney U test, a non-parametric equivalent to the *t*-test, is performed between the mean ± SD of all features across the PE-TE and PL-TL groups [42]. Those features with a significant value below 0.05 are used for further analysis. This way, the redundant and non-discriminative features, which can reduce the power of prediction of classifier and increase the training time, are discarded.

### 3.6. Data Segmentation

After reducing dimension of the feature vector, the remaining ones are normalized between 0 and 1 and split into training–validation (70%) and testing (30%) subsets. The training–validation is performed using a stratified 10-fold cross-validation. Nonetheless, due to the employment of data balancing algorithm that generates artificial preterm data, random sampling for generation of training–validation and testing subsets may lead to misleading results [27,43,47,49]. For instance, if a classifier achieves a sensitivity of 90% for the prediction of preterm labor after data balancing, it is not clear how much of that wrongly classified 10% is considered to be real data. Indeed, if such an error is highly correlated with the real EHG data, not the synthetically generated ones, the performance may not be reliable. To overcome this issue, we have used only the generated preterm data for training–validation step and used the real preterm data for testing. This way, the results are more realistic and the reliability of the proposed method can be validated better.

### 3.7. Classification

In order to classify between term and preterm labor, we employ four classifiers, i.e., AdaBoost, support vector machine (SVM), decision tree (DT), and random forest (RF), which have different learning strategies. The reason for employing the mentioned classifiers is that their reliability and robustness for the discrimination between the preterm and term cases was proven by other authors [37,42,50,51]. The hyperparameters of each classifier are optimized during the training–validation process using the Bayesian optimization method [52]. In order to assess the performance of each classifier, sensitivity (Sen), specificity (Spe), accuracy (Acc), and area under the curve (AUC) are calculated as follows:(9)$$\mathrm{Sen}=\frac{T_{P}}{T_{P}+F_{N}}\times 100,$$
(10)$$\mathrm{Spe}=\frac{T_{N}}{T_{N}+F_{P}}\times 100,$$
(11)$$\mathrm{Acc}=\frac{T_{P}+T_{N}}{T_{P}+T_{N}+F_{N}+F_{P}}\times 100,$$
(12)$$\mathrm{AUC}=\int \mathrm{Sen}\left(T\right){\left(1-\mathrm{Spe}\right)}^{\prime }\left(T\right)dT,$$
where $T_{P}$ and $F_{N}$ represent the number of correctly and wrongly classified preterm cases, $T_{N}$ and $F_{P}$ stand for the number of correctly and wrongly classified term cases, and *T* is the binary threshold of the classifier.

## 4. Results and Discussion

Addressing preterm prediction from an engineering point of view can be a challenging task as there is a gap between medical sciences and mathematics. In this paper, we aimed to extract features from EHG signals which are related to a physiologically justifiable expectation that more intense and frequent uterus contractions near the delivery will happen. The extracted features were a combination of measures that represent the EHG signal’s amplitude, energy, and complexity. On one hand, more intense and frequent uterus contractions can influence the amplitude and energy of the signal. On the other hand, such a phenomenon can also lead to more complexity of the signal.

### 4.1. Selected Features

Table 2 displays the selected features for both groups after conducting the Whitney U test. As it can be seen, most of the selected features for both groups belonged to the CH1. Moreover, regardless of the channels, most of the features were selected from ${IMF}_{1}$ and $IMF_{2}$. In summary, 35 and 36 features were selected for the PE-TE and PL-TL groups, respectively.

### 4.2. Classification Results

As already stated, simple random sampling for segmenting the data into training–validating and testing sets might lead to a misleading classification result as it is not clear how much the classifier is sensitive to the real preterm EHG data. To overcome this issue, features related to real EHG data were separated from the feature vector and were only used for the testing. Table 3 displays the classification results on the unseen testing dataset for all classifiers.

As displayed, the best classification results for the PE-TE group were obtained by the AdaBoost classifier, with a mean Acc of 95%, Sen of 92%, Spe of 97%, and AUC of 0.99 followed by the RF classifier with a mean Acc of 92%, Sen of 86%, Spe of 96%, and AUC of 0.97. Regarding the PL-TL group, the AdaBoost classifier again achieved the highest mean Acc of 93%, Sen of 90%, Spe of 94%, and AUC of 0.98, followed by RF with a mean Acc of 90%, Sen of 84%, Spe of 96%, and AUC of 0.95. Indeed, AdaBoost outperformed the other classifiers for both the PE-TE and PL-TL groups. According to the carried out *t*-test statistical analysis, there was a significant difference between the Sen values obtained by AdaBoost and the other classifiers for both groups, indicating better performance of AdaBoost to predict the preterm labor. Yet, although slightly better classification results were obtained for the PE-TE group, no significant difference was found between the classification results of the two groups (*p* > 0.05). In order to make sure that our statistical analysis did not incorrectly reject the null hypothesis, we also employed the Bonferroni–Holm correction for multiple comparisons, which confirmed the primary results. Figure 4 shows the receiver operating characteristic (ROC) curves obtained by each classifier for both groups.

### 4.3. Sensitivity of Classifiers for Preterm Labor Based on Only Real EHG Data

Although the reported results in the previous section are somehow more realistic as all the real EHG data related to the preterm labor were employed for the unseen testing, the reported Sen values are based on the combination of both real and synthesized EHG preterm data. To this aim, we also computed the Sen of each classifier based on only real preterm EHG data (Figure 5). For both groups, the AdaBoost classifier achieved a mean Sen of almost 89% which means at least 17 of 19 preterm cases were identified correctly.

### 4.4. Comparison against the State-of-the-Art Methods

Table 4 compares the results obtained by the proposed method to the state of the art in terms of the Acc, Sen, Spe, and AUC. It should be noted that we only considered studies that addressed such a prediction based on different pregnancy weeks.

Although some studies achieved a higher accuracy than ours, they employed features that required parameter setting before computation, e.g., sample entropy. On the other hand, accurate parameter setting of such features plays an important role for their performance [9]. In addition, as such regulation is performed experimentally, it is not clear how well such features work for a new set of data. On the contrary, the nonlinear features employed in our study were parameter free. Furthermore, the reported Sen values in our study are more realistic as all real preterm cases were unseen in the training–validating process and used solely for testing.

### 4.5. Future Work

Although the proposed algorithm showed promising results, the following issues should be addressed in future work. First, categorical features, e.g., age, height, and weight, can be employed as extra information for classification. Second, reducing the number of employed channels, e.g., a single EHG channel, should be investigated as welcome in long-term home-based pregnancy monitoring systems. Third, the effectiveness of the proposed method should be further investigated using different versions of filtered EHG signals (e.g., 0.3 to 3 Hz) or even with different frequency ranges as suggested in [53]. Fourth, though the selected features showed acceptable performance, it should be noted nonetheless that the employed strategy might not exclude the redundant features. Moreover, those features that were considered non-discriminative may show a better performance when combined together. Therefore, a more advanced feature selection method can be investigated for future work. Fifth, it has been shown that the continuous wavelet transform is a promising tool for analyzing EHG data [53]; thus, it can also be used to decompose EHG signals instead of EMD. Last, we have not considered isolating bursts from the EHG to predict preterm labor. Isolating bursts from the EHG may lead to more accurate prediction of preterm labor, yet it requires the supervision of qualified personnel or the simultaneous use of a tocodynamometer.

## 5. Conclusions

This paper presents a new method for the prediction of preterm labor using EHG signals from different pregnancy weeks. The similarity between the obtained results for both the PE-TE and PL-TL groups confirms the reliability of the proposed method for the prediction of preterm labor. More importantly, the proposed method showed a high mean of sensitivity to the real preterm EHG data in both groups.

## Figures and Tables

**Figure 1 sensors-23-05965-f001:**
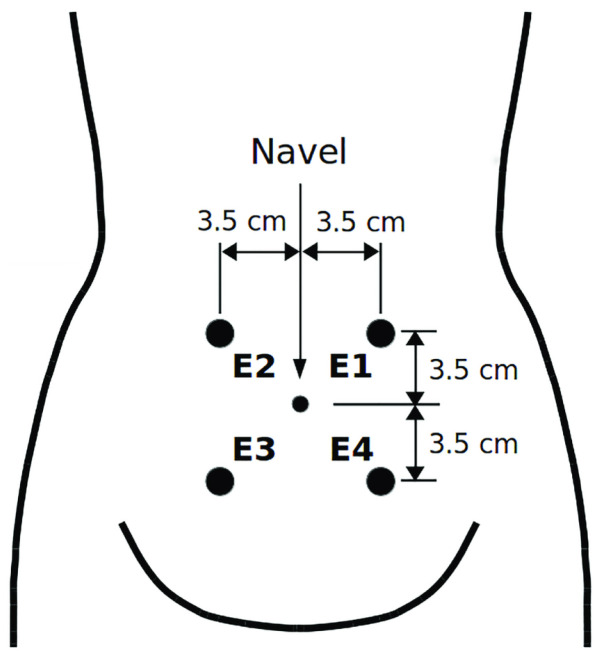
The placement of EHG electrodes, adopted from [35].

**Figure 2 sensors-23-05965-f002:**
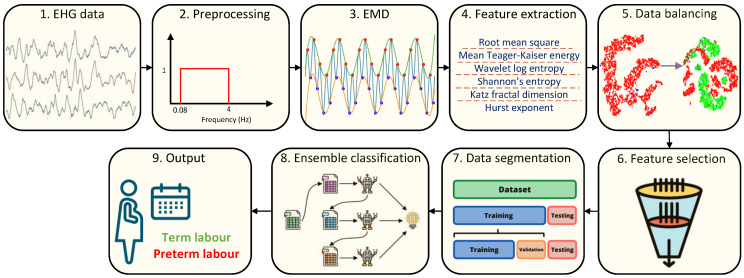
The block diagram of the proposed method for the prediction of preterm labor using EHG signals.

**Figure 3 sensors-23-05965-f003:**
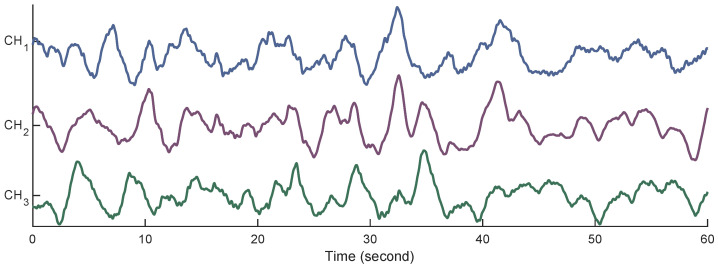
Examples of the filtered EHG signals from all three channels.

**Figure 4 sensors-23-05965-f004:**
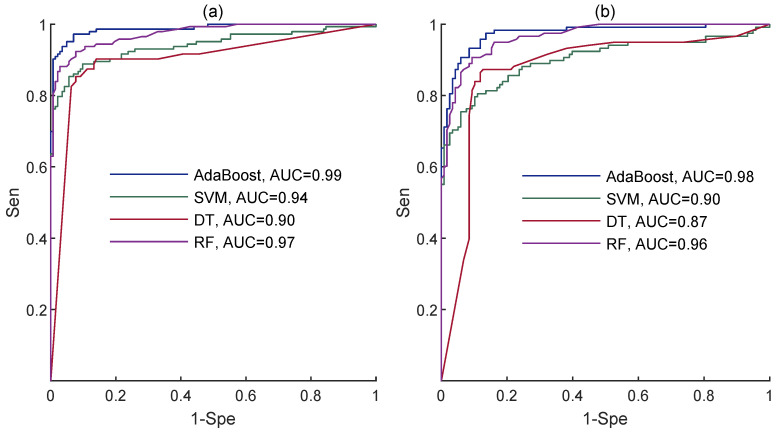
The ROC curves of all classifiers for (**a**) PE-TE and (**b**) PL-TL groups.

**Figure 5 sensors-23-05965-f005:**
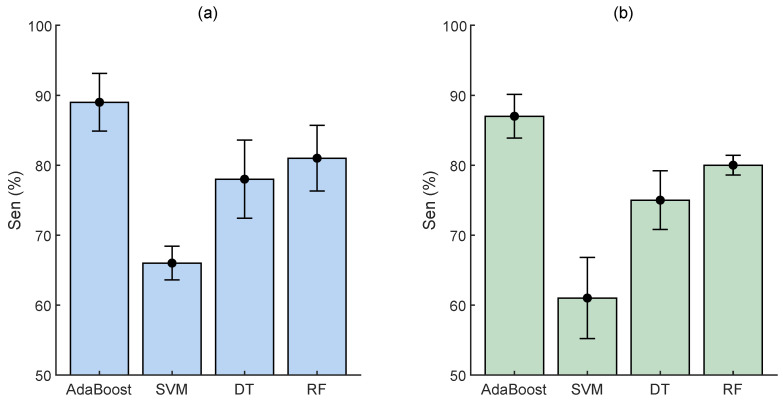
Sensitivity of all classifiers for real preterm EHG data. (**a**) PE-TE and (**b**) PL-TL groups.

**Table 1 sensors-23-05965-t001:** The number of term and preterm deliveries based on the pregnancy week.

Gestational Time of Recording	Delivery Type	
	Preterm	Term
Before 26th week	Preterm Early (PE), n = 19	Term Early (TE), n = 143
After 26th week	Preterm Later (PL), n = 19	Term Later (TL), n = 119

**Table 2 sensors-23-05965-t002:** Selected features for both groups.

PE-TE Group											
${\mathrm{CH}}_{1}$				${\mathrm{CH}}_{2}$				${\mathrm{CH}}_{3}$			
${\mathrm{IMF}}_{1}$	${\mathrm{IMF}}_{2}$	${\mathrm{IMF}}_{3}$	${\mathrm{IMF}}_{4}$	${\mathrm{IMF}}_{1}$	${\mathrm{IMF}}_{2}$	${\mathrm{IMF}}_{3}$	${\mathrm{IMF}}_{4}$	${\mathrm{IMF}}_{1}$	${\mathrm{IMF}}_{2}$	${\mathrm{IMF}}_{3}$	${\mathrm{IMF}}_{4}$
RMS	RMS	WLE	WLE	RMS	WLE	WLE	RMS	RMS	RMS	RMS	WLE
MTKE	MTKE	SE	SE	MTKE	SE	KFD	KFD	MTKE	KFD	WLE	SE
WLE	WLE	HE	KFD	SE			HE	SE	HE		
SE	KFD		HE								
PL-TL group											
${\mathrm{CH}}_{1}$				${\mathrm{CH}}_{2}$				${\mathrm{CH}}_{3}$			
${\mathrm{IMF}}_{1}$	${\mathrm{IMF}}_{2}$	${\mathrm{IMF}}_{3}$	${\mathrm{IMF}}_{4}$	${\mathrm{IMF}}_{1}$	${\mathrm{IMF}}_{2}$	${\mathrm{IMF}}_{3}$	${\mathrm{IMF}}_{4}$	${\mathrm{IMF}}_{1}$	${\mathrm{IMF}}_{2}$	${\mathrm{IMF}}_{3}$	${\mathrm{IMF}}_{4}$
RMS	RMS	WLE	WLE	RMS	RMS	WLE	RMS	RMS	RMS	RMS	WLE
MTKE	MTKE	SE	SE	KFD	KFD	HE	SE	WLE	MTKE	HE	KFD
WLE	WLE	KFD	KFD	SE	HE			HE	KFD		
SE	HE		HE								
HE											

**Table 3 sensors-23-05965-t003:** Classification results on the unseen testing subset for both groups using all classifiers.

Group	PE-TE				PL-TL			
Classifier	Sen	Spe	Acc	AUC	Sen	Spe	Acc	AUC
AdaBoost	92%	97%	95%	0.99	90%	94%	93%	0.98
SVM	66%	99%	83%	0.93	64%	98%	81%	0.89
DT	85%	92%	88%	0.90	85%	87%	86%	0.86
RF	86%	96%	92%	0.97	84%	96%	90%	0.95

**Table 4 sensors-23-05965-t004:** The comparison of our study with state-of-the-art methods.

Study	Group and No. of Data	Classifier	Acc	Sen	Spe	AUC
Ours	PE (n = 143)-TE (n = 143)	AdaBoost	95%	92%	97%	0.99
	PL (n = 119)-TL (n = 119)		93%	90%	93%	0.98
[42]	PE (n = 135)-TE (n = 143)	RF	92%	88%	96%	0.88
	PL (n = 111)-TL (n = 119)		93%	89%	97%	0.80
[43]	PE (n = 93)-TE (n = 93)	QDA	97%	100%	95%	N.A
	PL (n = 57)-TL (n = 57)		100%	100%	100%	N.A
[35]	PE (n = 140)-TE (n = 143)	QDA	100%	100%	100%	1.0
[44]	PE-TE, n is not reported.	SVM	96.5%	94%	99%	0.99
	PL-TL, n is not reported.		92.5%	88%	97%	0.98

## Data Availability

This database is publicly available at https://physionet.org/content/tpehgdb/1.0.1/, accessed on 1 October 2022.

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
