# Peer review of "Prediction of Preterm Labor from the Electrohysterogram Signals Based on Different Gestational Weeks"

_sensors, 2023, doi:10.3390/s23135965_

Round 1

Reviewer 1 Report

This paper proposed a method for reliable prediction of preterm labor from the EHG signals based on different pregnancy weeks. Several parameter-free linear and nonlinear features from the EHG signals were extracted and then fed to several classifiers for the final prediction.

The results are reasonable.

1) Can the authors explain the results in Table 3, Why the classifier performance for PE-TE is better than PL-TL?  Generally, it is relatively easy to discriminate the preterm labor and term labor when close to delivery?

2) The authors selected features using significant value from statistical analysis. Can this method eliminate the redundant features?

Besides, even some features themselves are non-discriminative between groups, however they may be discriminative combing together.

The features selection method could be improved.

3) The features normalized for DT and RF may lead to information lost.

4Any explanation for selection the first four decomposed IMFs for the feature extraction?

Author Response

Reviewer I:

This paper proposed a method for reliable prediction of preterm labor from the EHG signals based on different pregnancy weeks. Several parameter-free linear and nonlinear features from the EHG signals were extracted and then fed to several classifiers for the final prediction.

The results are reasonable.

Author response:

Thanks for your comments. We have addressed all issues raised by you. 

1) Can the authors explain the results in Table 3, Why the classifier performance for PE-TE is better than PL-TL?  Generally, it is relatively easy to discriminate the preterm labor and term labor when close to delivery?

Author response:

Thanks for your comment. Firstly, please note that there is no noticeable difference between the classification results for both groups by all classifiers. Furthermore, as shown in Table IV, some studies that achieved higher accuracy for the PL-TL group have used different sample size. To address your comment properly, we have conducted the statistical test between the obtained results and added the following sentence to the results section of our paper:

Yet, although slightly better classification results were obtained for the PE-TE group, no significant difference was found between the classification results of two groups (P>0.05).

2) The authors selected features using significant value from statistical analysis. Can this method eliminate the redundant features?

Besides, even some features themselves are non-discriminative between groups, however they may be discriminative combing together.

The features selection method could be improved.

 Author response:

Thanks for your comment. We believe the reviewer raised a fair and correct comment. Please be noted that we have the same strategy for the feature extraction as suggested in [42]. To address your comment, we have added the following sentence to our future work:

Fourthly, though the selected features showed acceptable performance, nonetheless, it should be noted that the employed strategy might not exclude the redundant features. Moreover, those features that were considered non-discriminative may show a better performance when combing together. Therefore, a more advance feature selection method can be investigated for the future work.

3) The features normalized for DT and RF may lead to information lost.

Author response:

Thanks for your comment. Please note that we used RF and DT for the classification problem, not regression. For classification tasks, the output of the RF or DT is the class selected by most trees. For regression tasks, the mean or average prediction of the individual trees is returned. Therefore, data normalization won’t affect the output for RF and DT classifiers while it will affect the output for their regressors.

4) Any explanation for selection the first four decomposed IMFs for the feature extraction?

Author response:

Thanks for your comment. Our hypothesis is that EHG signals related to preterm class show more frequent and stronger contractions, i.e., contain more higher frequency components, thus, when employing EMD which decomposes a signal from high to low frequency components, it might be better to only consider those IMFs that have a higher frequency range for discrimination of term and preterm classes. To make it clearer, we have revised the following statement in the last paragraph of our introduction

As the uterus contractions become more intense and frequent near the delivery, it can be expected that the EHG signal indicating preterm labor will exhibit stronger and more frequent contractions compared to term labor. In other words, the EHG signal associated with the preterm class will have a higher presence of high-frequency components. On the other hand, the EMD technique breaks down a signal into its frequency components from high to low, so we utilize only the initial four IMFs the feature extraction.

Reviewer 2 Report

minor language corrections are needed

minor language corrections are needed

Author Response

Reviewer II:

1) minor language corrections are needed.

Author response:

Thanks for your comment. We have revised and checked the English grammar accurately.   

Reviewer 3 Report

1. The lack of a relevant reference for selecting the band-pass filter range raises concerns about the validity and appropriateness of the chosen frequency range.

2. Justification is needed for why only the first four decomposed IMFs were chosen for feature extraction. It is unclear why other IMFs were excluded and whether this choice affected the analysis.

3. The article does not adequately explain why the EMD procedure was performed and how it contributes to the preprocessing or processing stage. Without this clarification, it is not easy to understand the rationale behind using EMD in the study.

4. It is unclear whether the analysis focused solely on EHG bursts or if the entire EHG-IMFs signals were considered. A justification for this choice is necessary to understand the scope and limitations of the analysis.

5. The selection of EHG features in the study appears arbitrary, and the article does not provide sufficient reasoning or evidence to support the specific feature choices.

6. The absence of a reference for WLE makes it challenging to assess its relevance and validity in the study context.

7. Due to the application of multiple comparisons in the study, the significance level should be adjusted using a method such as the Bonferroni correction to minimize the chances of false positives.

8. The number of EHG-IMF segments analyzed after applying the balancing (SASYNO) method is not mentioned, making it difficult to evaluate the impact and effectiveness of this method.

9. A recent study demonstrating the potential enhancement of preterm-term birth classification using continuous wavelet transform and entropy-based methods on electrohysterogram signals strengthens the argument for considering alternative approaches and techniques.

10. The article lacks a clear and explicit statement regarding the physiological interpretation of the results, making it challenging to understand the clinical implications and significance of the findings.

Author Response

Reviewer III:

1) The lack of a relevant reference for selecting the band-pass filter range raises concerns about the validity and appropriateness of the chosen frequency range.

Author response:

Thanks for your comment. Please note that the frequency range of filtering for the pre-processing has been suggested by the authors who developed this database. Indeed, when downloading the public database, each signal was digitally filtered using 3 different 4-pole digital Butterworth filters with a double-pass filtering scheme. The band-pass cut-off frequencies were:

from 0.08Hz to 4Hz;

from 0.3Hz to 3Hz;

from 0.3Hz to 4Hz.

Thus, we have only used the filtered data from the database (0.08Hz to 4Hz). Nonetheless, we understand the reviewer concern so we have added the following statement to our future work:

Thirdly, the effectiveness of proposed method should be further investigated using different version of filtered EHG signals (e.g., 0.3 to 3 Hz) or even with different frequency ranges as suggested in [53].

2) Justification is needed for why only the first four decomposed IMFs were chosen for feature extraction. It is unclear why other IMFs were excluded and whether this choice affected the analysis.

Author response:

Thanks for your comment. Our hypothesis is that EHG signals related to preterm class show more frequent and stronger contractions, i.e., contain more higher frequency components, thus, when employing EMD which decomposes a signal from high to low frequency components, it might be better to only consider those IMFs that have a higher frequency range for discrimination of term and preterm classes. To make it clearer, we have revised the following statement in the last paragraph of our introduction:  

As the uterus contractions become more intense and frequent near the delivery, it can be expected that the EHG signal indicating preterm labor will exhibit stronger and more frequent contractions compared to term labor. In other words, the EHG signal associated with the preterm class will have a higher presence of high-frequency components. On the other hand, the EMD technique breaks down a signal into its frequency components from high to low, so we utilize only the initial four IMFs the feature extraction.

3) The article does not adequately explain why the EMD procedure was performed and how it contributes to the preprocessing or processing stage. Without this clarification, it is not easy to understand the rationale behind using EMD in the study.

Author response:

Thanks for your comment. As we answered to your previous comment, we hypothesized that EHG signal related to preterm labor contains more high frequency components. Thus, instead of using the EHG signal, employing a method that decomposes the signal into different frequency levels and using levels with a higher frequency range can be a better option. To address your comment properly, we have revised the following statement in the last paragraph of our introduction:

The motivation for using EMD over the other decomposition methods like wavelet is that it does not require a pre-defined basis function. Furthermore, EMD does not assume any prior knowledge about the signal, such as stationarity or linearity. This characteristic makes EMD particularly useful when dealing with nonstationary signals such as EHG that often exhibit complex and unpredictable behavior.  

4) It is unclear whether the analysis focused solely on EHG bursts or if the entire EHG-IMFs signals were considered. A justification for this choice is necessary to understand the scope and limitations of the analysis.

Author response:

Thanks for your comment. We have only used entire IMF signals with windowing. The burst detection should be either done manually, which is against the nature of reducing human intervention, or should be done based on an algorithm that has been developed for this purpose, which is out of scope of this research. To address your comment, we have added the following statement to our future work:

Lastly, we have not considered isolating bursts from the EHG to predict preterm labor. Isolating bursts from the EHG may lead to more accurate prediction of preterm labor, yet, it requires the supervision of qualified personnel or the simultaneous use of TOCO.

5) The selection of EHG features in the study appears arbitrary, and the article does not provide sufficient reasoning or evidence to support the specific feature choices.

Author response:

Thanks for your comment. Please note that we have used features that are parameter free and need no tuning before the computation. To address your concern, we have added the following section in the discussion of our paper where we also address the comment 10 raised by you as follows:

Addressing preterm prediction from an engineering point of view can be a challenging task as there is a gap between medical sciences and mathematics. In this paper, we aimed to extract features from EHG signals which are related to a physiologically-justifiable expectation that more intense and frequent uterus contractions near the delivery will happen. The extracted features were a combination of measures that represent EHG signal’s amplitude, energy, and complexity. On one hand, more intense and frequent uterus contractions can influence the amplitude and energy of the signal. On the other hand, such phenomenon can also lead to more complexity of the signal.     

6) The absence of a reference for WLE makes it challenging to assess its relevance and validity in the study context.

Author response:

Thanks for your comment. We have added a proper reference for the WLE as follows:

Here, we used WLE and SE defined as follows [46]:

7) Due to the application of multiple comparisons in the study, the significance level should be adjusted using a method such as the Bonferroni correction to minimize the chances of false positives.

Author response:

Thanks for your comment. We have considered your comment and applied Bonferroni-Holm methods for the Spe, Sen, and Acc values obtained by different classifiers and added the following sentence to our classification results section:

In order to make sure that our statistical analysis has not incorrectly rejected the null hypothesis, we have also employed the Bonferroni-Holm correction for multiple comparisons, which confirmed the primary results.

8) The number of EHG-IMF segments analyzed after applying the balancing (SASYNO) method is not mentioned, making it difficult to evaluate the impact and effectiveness of this method.

Author response:

Thanks for your comment. Please note that the data balancing is performed on the extracted features, not on the IMFs. Thus, data balancing has no impact on the decomposition of the signals. 

9) A recent study demonstrating the potential enhancement of preterm-term birth classification using continuous wavelet transform and entropy-based methods on electrohysterogram signals strengthens the argument for considering alternative approaches and techniques.

Author response:

Thanks for your comment. We have considered your suggestion and added one sentence to our future work with a proper reference as follows:

Fifthly, it has been shown that the continuous wavelet transform is a promising tool for analyzing EHG data [53], thus, it can be also used to decompose EHG signals instead of EMD.

10) The article lacks a clear and explicit statement regarding the physiological interpretation of the results, making it challenging to understand the clinical implications and significance of the findings.

Author response:

Thanks for your comment. We have added the following sentence to our discussion to address your comment.

Addressing preterm prediction from an engineering point of view can be a challenging task as there is a gap between medical sciences and mathematics. In this paper, we aimed to extract features from EHG signals which are related to a physiologically-justifiable expectation that more intense and frequent uterus contractions near the delivery will happen.

Reviewer 4 Report

- The novelty and merit of this study is not clear. 

- There are several studies and methods that have applied to the same dataset. 

- The dataset used in this study has been extensively examined. Several excellent results were reported. 

- Many relevant studies were not included and discussed in this study. 

Author Response

Reviewer IV:

1) The novelty and merit of this study is not clear. 

Author response:

Thanks for your comment. We believe the reviewer raised a fair concern. To address your comment, we have revised the last paragraph of our introduction and some part of our discussion where we have clearly stated what the limitation of the previous studies are and what our solution is as follows:

Introduction:

Despite the promising results for the prediction of preterm labor using PE-TE and PL-TL groups, there are two issues that have not been addressed in the aforementioned studies. Firstly, the employment of nonlinear features such as sample entropy and fuzzy entropy, which require parameters setting before computation, may threaten the generality of the method as such tuning is performed experimentally and it is not clear how well it works for unseen data. Secondly, with artificially balanced dataset the reported sensitivity of methods to the real preterm EHG data was not accurately investigated, i.e., in case of sensitivity of 90%, it is not clear how much of that missed 10% are related to the real preterm data. The objective of this paper is to present a reliable method for predicting the preterm labor based on different pregnancy weeks. For this aim, we extract several parameter-free linear and nonlinear features from the EHG signals decomposed by EMD and then feed them to several classifiers for the final prediction. As the uterus contractions become more intense and frequent near the delivery, it can be expected that the EHG signal indicating preterm labor will exhibit stronger and more frequent contractions compared to term labor. In other words, the EHG signal associated with the preterm class will have a higher presence of high-frequency components. On the other hand, the EMD technique breaks down a signal into its frequency components from high to low, so we utilize only the initial four IMFs for the feature extraction.

Discussion:

Table 4 compares the results obtained by the proposed method to the state-of the-art in terms of Acc, Sen, Spe, and AUC. It should be noted that we have only considered studies that addressed such prediction based on different pregnancy weeks. Although some studies achieved a higher accuracy than ours, yet, they have employed features that required parameters setting before computation, e.g., sample entropy. On the other hand, accurate parameter setting of such features plays an important role for their performance [9]. In addition, as such regulation is performed experimentally, it is not clear how well such features work for a new set of data. In contrary, the nonlinear features employed in our study are parameter-free. Furthermore, the reported Sen values in our study is more realistic as all real preterm cases were unseen in training-validating process and used solely for testing.

2) There are several studies and methods that have applied to the same dataset. 

Author response:

Thanks for your comment. We agree with the reviewer that this database has been examined by several studies. Yet, the discrimination of preterm prediction based on different gestational weeks has been less investigated. Thus, we have only considered research that addressed this issue directly.  

3) The dataset used in this study has been extensively examined. Several excellent results were reported. 

Author response:

Thanks for your comment. Please refer to our answer to your last comment.

4) Many relevant studies were not included and discussed in this study. 

Author response:

Thanks for your comment. As we answered to your first comment, we mostly considered the studies that have addressed the discrimination of preterm prediction based on different gestational weeks. To the best of our knowledge, we have mentioned all the studies that investigated this issue. Yet, we would be grateful if the reviewer can purpose other proper references for our paper.   

Round 2

Reviewer 1 Report

the authors give  responses to all the comments.

Reviewer 3 Report

The authors have addressed all my concerns.